# Trends in Influenza Vaccination Rates among a Medicaid Population from 2016 to 2021

**DOI:** 10.3390/vaccines11111712

**Published:** 2023-11-11

**Authors:** Behzad Naderalvojoud, Nilpa D. Shah, Jane N. Mutanga, Artur Belov, Rebecca Staiger, Jonathan H. Chen, Barbee Whitaker, Tina Hernandez-Boussard

**Affiliations:** 1Department of Medicine, Stanford University, Stanford, CA 94305, USA; behzadn@stanford.edu (B.N.); rstaiger@stanford.edu (R.S.);; 2Stanford Center for Biomedical Informatics Research, Stanford, CA 94305, USA; 3Center for Biologics Evaluation and Research, Office of Biostatistics and Pharmacovigilance, U.S. Food and Drug Administration, Silver Spring, MD 20993, USA; namangolwa.mutanga@fda.hhs.gov (J.N.M.);; 4Division of Hospital Medicine, Stanford, CA 94305, USA; 5Clinical Excellence Research Center, Stanford, CA 94304, USA; 6Department of Surgery, Stanford University School of Medicine, Stanford, CA 94305, USA; 7Department of Biomedical Data Science, Stanford University, Stanford, CA 94305, USA

**Keywords:** influenza vaccine, vaccine, Medicaid, COVID-19, vaccination uptake

## Abstract

Seasonal influenza is a leading cause of death in the U.S., causing significant morbidity, mortality, and economic burden. Despite the proven efficacy of vaccinations, rates remain notably low, especially among Medicaid enrollees. Leveraging Medicaid claims data, this study characterizes influenza vaccination rates among Medicaid enrollees and aims to elucidate factors influencing vaccine uptake, providing insights that might also be applicable to other vaccine-preventable diseases, including COVID-19. This study used Medicaid claims data from nine U.S. states (2016–2021], encompassing three types of claims: fee-for-service, major Medicaid managed care plan, and combined. We included Medicaid enrollees who had an in-person healthcare encounter during an influenza season in this period, excluding those under 6 months of age, over 65 years, or having telehealth-only encounters. Vaccination was the primary outcome, with secondary outcomes involving in-person healthcare encounters. Chi-square tests, multivariable logistic regression, and Fisher’s exact test were utilized for statistical analysis. A total of 20,868,910 enrollees with at least one healthcare encounter in at least one influenza season were included in the study population between 2016 and 2021. Overall, 15% (N = 3,050,471) of enrollees received an influenza vaccine between 2016 and 2021. During peri-COVID periods, there was an increase in vaccination rates among enrollees compared to pre-COVID periods, from 14% to 16%. Children had the highest influenza vaccination rates among all age groups at 29%, whereas only 17% were of 5–17 years, and 10% were of the 18–64 years were vaccinated. We observed differences in the likelihood of receiving the influenza vaccine among enrollees based on their health conditions and medical encounters. In a study of Medicaid enrollees across nine states, 15% received an influenza vaccine from July 2016 to June 2021. Vaccination rates rose annually, peaking during peri-COVID seasons. The highest uptake was among children (6 months–4 years), and the lowest was in adults (18–64 years). Female gender, urban residency, and Medicaid-managed care affiliation positively influenced uptake. However, mental health and substance abuse disorders decreased the likelihood. This study, reliant on Medicaid claims data, underscores the need for outreach services.

## 1. Introduction

Seasonal influenza continues to be a significant cause of morbidity and mortality and was the ninth leading cause of death in the United States in 2019 and 2020 [1,2]. The incidence of influenza in the U.S. is estimated to be 8.3% for all ages, 9.3% for those younger than 18 years, and 8.9% for adults 18–64 years during the flu seasons between 2010 and 2016 [3]. The Centers for Disease Control and Prevention (CDC) estimates that between 2016 and 2020, influenza resulted in 29–35 million illnesses, 380,000–710,000 hospitalizations, and 20,000–52,000 deaths annually in the U.S. [4]. The average annual total economic burden to the healthcare system and society due to influenza (e.g., doctors’ visits, hospitalization, and lost income) is estimated at USD 11.2 billion (ranging from USD 6.3 to 25.3 billion) [5].

Vaccination is a safe and effective intervention for preventing illness, hospitalizations, and death due to influenza and other vaccine-preventable diseases [6,7]. The CDC recommends influenza vaccination every season for persons aged six months and older, including at-risk groups such as people with chronic diseases, including those with asthma, heart disease, and diabetes [8]. It has been reported that influenza vaccination is associated with an estimated 41% reduction in the risk of hospitalized influenza illness among older individuals [9], an estimated 15% reduction in hospital admissions for pneumonia or influenza among people with diabetes [10], and an estimated 40% reduction in hospitalizations among pregnant women [11]. Vaccination against influenza has been demonstrated to be 91.5% effective at preventing influenza-related hospitalizations among infants within the first 6 months of life when their mothers were vaccinated during pregnancy [12]. Hence, vaccination against influenza has the potential to significantly reduce severe clinical outcomes that contribute to social, economic, and healthcare-related burdens.

The Centers for Disease Control and Prevention (CDC) Advisory Committee on Immunization Practices (ACIP) recommends using routine healthcare visits and encounters as an important opportunity to provide health maintenance interventions, including administering an influenza vaccine [13]. Despite this recommendation and the availability of influenza vaccine as a preventive service, vaccination rates remain low among Medicaid enrollees, who are considered the most economically and socially vulnerable population in the U.S. [7]. Medicaid, a federally funded healthcare program in the United States provides health coverage to eligible low-income adults, children, pregnant people, elderly adults, and people with disabilities [14]. Additionally, Medicaid serves other vulnerable groups, such as foster children and refugees, offering them essential access to healthcare services.

Influenza vaccination rates among Medicaid adults are lower than privately insured individuals (32.8% vs. 40.8%, respectively) but almost twice as high as among the uninsured population, for which only 16.3% received a vaccination [7]. These rates are all well below the Healthy People 2030 goal of increasing the influenza vaccination rate to 70% [15]. Therefore, increasing influenza vaccination rates among the Medicaid population is a public health priority that can be informed by exploring the factors associated with vaccine uptake and the differences between subpopulations [4].

As of May 2023, over 87 million unduplicated individuals have been enrolled in Medicaid since the start of the program [16]. Large real-world data (RWD) sources, like Medicaid administrative claims data, arise from regular healthcare delivery, administration, and reimbursement processes. These data offer opportunities to explore population-level health studies and assist in creating guidelines and decision-making tools for clinical practice [17,18]. The U.S. Food and Drug Administration (FDA) has also highlighted the need for improved RWD use to generate real-world evidence (RWE) to enhance product safety and surveillance efforts, including for licensed vaccine products such as the influenza vaccine [18,19].

The objective of this work is to estimate influenza vaccination rates among Medicaid enrollees using Medicaid claims data. Previous studies have been conducted to evaluate influenza vaccine uptake among individuals with pre-existing medical conditions including rheumatic or malignant diseases, sickle cell disease, and chronic kidney or liver disease, but few studies have been conducted among pregnant persons or individuals with diabetes or other comorbidities who make up a large proportion of the Medicaid population [20,21,22]. We hypothesize that people with at-risk conditions (i.e., diabetes, pregnancy) are more likely to get vaccinated. Though this study focuses on influenza vaccination rates with limited years of available data, the insights from such analyses may be applicable to the timely understanding of factors influencing the uptake of other vaccines for preventable diseases, including COVID-19.

## 2. Materials and Methods

### 2.1. Data Source

This was a cross-sectional study utilizing Medicaid claims data collected between 2016 and 2021 from nine U.S. states—three western states, three midwestern states, and three southern states. Of the nine states, three states reported claims for the subset of fee-for-service enrollees, three reported claims from one (major) Medicaid managed care plan, and three reported claims for all Medicaid enrollees (fee-for-service and managed care enrollees). These data included information regarding medical claims, patient enrollment and demographics, pharmacy claims, and provider information. This study was approved by Stanford University’s institutional review board.

### 2.2. Population

The study population was composed of Medicaid enrollees living in nine U.S. states who had at least one healthcare encounter during at least one influenza season within the study period. Enrollees were excluded if they were less than 6 months of age, 65 years and older, or did not have an in-person encounter (i.e., telehealth only) since these encounters did not provide an in-person opportunity for vaccine administration.

### 2.3. Measures

An influenza season was defined as the period between July of one year to June of the following year. We divided the season into four sub-seasons: August–October, November–January, February–April, and May–July. However, vaccination rates during the initial and final months of each season (July and June) were extremely low. We utilized these two months, along with May, in the last subgroup (May–July) to better illustrate the segmentation of vaccine uptake across sub-seasons. To calculate the vaccination rate for each season, we divided the total number of vaccinated enrollees with at least one vaccination claim (as shown in Appendix A) by the total number of enrollees who have enrolled in or have valid Medicaid coverage and at least one healthcare encounter during the season.

We examined the trends in vaccination rates before and during the COVID-19 pandemic. A pre-COVID-19 group was created by summing the total number of enrollees within each of the four seasons between July 2016 and June 2020, and a peri-COVID-19 group comprised enrollees within the season between July 2020 and June 2021. The overall group consists of the total number of enrollees within each of the five seasons during the study period. To calculate the vaccination rate for a COVID-19 era, the total vaccinated enrollees and total enrollees were obtained by adding those values during the included seasons.

Patient-level variables included in this study were age, gender, and at-risk conditions, including diabetes and pregnancy status (see Table 1). Age groups were categorized as 6 months–4 years, 5–17 years, and 18–64 years. These age groups were developed based on the CDC recommendations for persons aged 6 months and older to receive an annual influenza vaccine [23,24]. Individuals aged 65 years and older were not included in this study as they are likely to be dually enrolled in Medicare, which would render their Medicaid claims an incomplete record of their total utilization. Age groups were constructed from the claims data based on the first time they received the influenza vaccine within the study period. Diabetes Mellitus (DM) diagnosis and pregnancy status were identified using the Current Procedural Terminology (CPT) system and International Classification of Diseases (ICD 9 and 10) codes as listed in Appendix A. Enrollees with a pregnancy-related ICD and CPT code within the seasons considered were classified as pregnant. We also used the Clinical Classification Software Redefined (CCSR), which aggregates ICD-10 Revision, Clinical Modification/Procedure Coding Systems (ICD-10 CM/PCS) codes into clinically meaningful categories [25]. The clinical categories with less than 2.0% prevalence within the study sample were not included in this study.

Census data were used to develop the Metropolitan and higher education variables. The classification of enrollees into metropolitan and non-metropolitan counties was generated using the enrollee’s zip code-level data and matched to the Rural–Urban Continuum Codes (RUCC) (Appendix A) that distinguish metropolitan counties by population size of the metro area and non-metropolitan county by its degree of urbanization and adjacency to a metro area [26]. Higher education status was generated using the county-level census data of the proportion of people with bachelor’s degrees or higher and the enrollee’s zip code. If an enrollee lived in an area with greater than the median percent of people with a bachelor’s degree or higher, the enrollee was categorized as living in an area with higher education; otherwise, enrollees were categorized as not living in an area with higher education.

### 2.4. Outcomes

The main outcome of interest was influenza vaccination rate, which was identified using enrollees’ influenza vaccination claims during the given season (see Appendix A). A secondary outcome was the number of in-person healthcare encounters (e.g., inpatient, emergency department, outpatient) in a given season. Telehealth, dental, and pharmacy-only encounters were excluded.

### 2.5. Statistical Analysis

We used Chi-square tests of independence to assess the difference in vaccination rates across different flu seasons. Multivariable logistic regression analysis was employed to estimate adjusted odds ratios (ORs) and their 95% confidence intervals (CIs) to identify potential factors associated with vaccine uptake. To this end, the Logit model was trained to predict enrollees’ seasonal vaccination based on the CCSR condition covariates having more than 2% frequency in both vaccinated and unvaccinated patients, as well as other binary covariates, including age groups, gender, living in a metropolitan area, living in a higher education area, and state with a managed care program. We utilized a searchable list of ICD-10-CM diagnosis codes mapped to CCSR categories, including default assignment for principal diagnosis, v2021.2 [27]. As a result, all codes found in the study cohort (including those in Appendix A) were mapped to the corresponding CCSR categories and incorporated into the logistic regression model. An analysis was conducted to examine the top 10 and bottom 10 odds ratios for CCSR categories. All analyses were performed in Python using the statsmodels library, *p* values ≤ 0.05 were considered to indicate statistical significance, and hypothesis tests were two-sided.

## 3. Results

### 3.1. Study Population Characteristics

A total of 20,868,910 enrollees with at least one healthcare encounter in at least one influenza season were included in the study population between 2016 and 2021 (Table 1). Overall, 56% (N = 11,597,177) of the study population were between the ages of 18 and 64, and 56% (N = 11,596,733) were female. A total of 80% (N = 16,686,225) of enrollees lived in a metropolitan area, 45% (N = 9,410,850) lived in an area with greater than median higher education, and 36% (N = 7,537,820) lived in a state with a managed care program. At least 16% (N = 3,348,282) of enrollees had diabetes (DM 1 or DM 2), and 16% (N = 3,423,662) of enrollees were pregnant.

### 3.2. Vaccination Rates during Pre-COVID and Peri-COVID Era

Table 1 shows the vaccination rates during the pre-COVID and peri-COVID eras. Total enrollees in this table indicate the total number of individuals who were enrolled in or had valid Medicaid coverage in the given COVID-19 era and had at least one healthcare encounter during each influenza season. Enrollees were counted separately per season and summed up to calculate the total number for the given COVID-19 era. The total vaccinated enrollees are the total of influenza vaccine claims made during the influenza seasons of the given COVID-19 era. Only one influenza vaccine claim was counted per enrollee during each influenza season. There were 81% (N = 16,807,523) enrollees during pre-COVID (2016–2020) and 19% (N = 4,061,387) during peri-COVID (2020–2021) periods who had at least one healthcare encounter during the given influenza season. Overall, 15% (N = 3,050,471) of enrollees received an influenza vaccine between 2016–2021. During peri-COVID periods, there was an increase in vaccination rates among enrollees compared to pre-COVID periods, from 14% to 16%.

From 2016 to 2021, 15% (N = 1,714,621) of females and 14% (N = 1,335,850) of males received vaccinations. During the peri-COVID period, we observed a slight increase in vaccination rates for enrollees. Vaccination rates among enrollees with any diabetes code (e.g., DM 1, DM 2, or unspecified diabetes) were 17% (N = 554,671). Enrollees living in a metropolitan area had a higher overall vaccination rate of 15% (N = 2,452,070) compared to enrollees not living in a metropolitan area at 14% (N = 598,401), and both had higher vaccination rates during the peri-COVID period compared to the pre-COVID period. A lower vaccination rate was observed for states that only had FFS during both pre-COVID and peri-COVID compared to Medicaid-managed care states.

Children aged 6 months–4 years had the highest influenza vaccination rates among all age groups (Table 1). For all seasons, the vaccination rates for each age group increased over time, except for the 5–17 years and 18–64 years group, which had a slight decrease in the July 2020–June 2021 season (Figure 1).

Appendix A shows the vaccination rates for all cohort characteristics throughout all influenza seasons. The differences between vaccination rates across influenza seasons were significant (*p* < 0.0001) using the Chi-square test.

Among the sub-seasons, most of the influenza vaccination uptake was during the months of August to October and November to January (Figure 2). In the overall group, 47% of those who were vaccinated received vaccines during August–October, and 41% of those vaccinated received vaccines during November–January (*p* < 0.05). A similar trend was seen when stratified for the pre- and peri-COVID era. As seen, the vaccination rates were extremely low during the initial and final months of each season (July, May, and June). Appendix A illustrates sub-season vaccination rates across all seasons, showing significant differences between sub-season vaccination rates (*p* < 0.05).

### 3.3. Average Number of In-Person Healthcare Encounters Per Influenza Seasons for Vaccinated, People with Diabetes, and Pregnant Women

The average number of in-person healthcare encounters stratified by vaccination status per influenza season for three groups of enrollees (overall population, diabetes, and pregnant enrollees) is shown in Figure 3. Enrollees who were vaccinated had a higher average number of healthcare encounters compared to those enrollees who were unvaccinated in each influenza season. Similar results were shown for enrollees with diabetes and enrollees who were pregnant. Over time, the average number of in-person healthcare encounters decreased for all groups considered for both vaccinated and unvaccinated. However, the last two seasons were during the COVID-19 pandemic.

### 3.4. Variables Associated with Influenza Vaccination Uptake

We investigated factors associated with receiving the influenza vaccine (Table 2). Within the compared age groups, the 6 months–4 years group had the highest odds of receiving an influenza vaccine compared to the 18–64 years (adult) age group for both the pre-COVID-19 and peri-COVID-19 eras. The 5–17 years age group had higher odds of getting influenza vaccination compared to the 18–64 years age group; however, the adjusted odds ratio for this group was slightly lower than the 6 months–4 years group. For both age groups, the odds of getting an influenza vaccine declined during the peri-COVID era compared to the pre-COVID era.

Males had lower odds of receiving the influenza vaccine compared to females for the overall study period. Males also had lower odds compared to females for both the pre- and peri-COVID era; however, odds were reduced further for the peri-COVID era. Enrollees living in an area with higher than median education had the same odds of receiving the influenza vaccine as those living in an area with lower than median education during the study period, as well as in the peri-COVID era. The odds for this group were slightly higher during the pre-COVID era compared to the group who lived in an area with lower than median education. The odds ratio during the peri-COVID era was the same for both metropolitan and non-metropolitan areas. Enrollees in a managed care program had higher odds of receiving the influenza vaccine compared to those not in a managed care program for the overall study period, pre-and peri-COVID eras.

All conditions associated with enrollees were mapped to 166 CCSR) categories in the logistic regression model. We investigated CCSR features that are most strongly associated with influenza vaccination uptake in the Medicaid population (Table 2). We observed that there were differences in the likelihood of receiving the influenza vaccine among enrollees based on their health conditions and medical encounters. Enrollees with certain conditions, such as heart failure and mental health disorders, were less likely to get vaccinated, while those with medical encounters, such as pregnancy-related issues and respiratory disorders, were more likely to receive the vaccine. The trends between pre- and peri-COVID eras were not consistent across the features considered as top 10 or bottom 10 based on the adjusted odds ratio, and the ORs varied between the time periods used for model fitting.

The adjusted odds ratios for two CCSR diabetic categories observed in the overall cohort study demonstrated a weak association between diabetes and vaccination uptake. In contrast, for most of the pregnancy CCSR categories, we observed a stronger association, indicating that pregnant enrollees were more likely to receive an influenza vaccination. Appendix A presents odds ratios for all CCSR features used in the logistic regression model, including patient number and vaccination rate.

## 4. Discussion

In this population-based study of Medicaid enrollees across nine states, 15% of enrollees with at least one in-person healthcare encounter received an influenza vaccine between July 2016 and June 2021. The number of Medicaid enrollees with an influenza vaccination increased each season, and rates were higher during the peri-COVID seasons compared to the pre-COVID seasons. Young children from 6 months to 4 years of age had the highest rates of influenza vaccination compared to other age groups. We found a strong association between the number of in-person encounters and the receipt of an influenza vaccine. Of note, several important factors related to mental health and substance abuse (i.e., schizophrenia, opioid-related, and tobacco-related disorders) were negatively associated with receipt of influenza vaccine. As overall vaccination rates sharply declined in the peri-COVID period, understanding which patient groups were at higher risk of not receiving the influenza vaccine might guide educational and outreach programs.

In this Medicaid population, the rates of influenza vaccination were lower than those reported in the Medicaid and CHIP Payment and Access Commission analysis [7]. However, CDC and Medicaid estimates were based on telephone surveys or personal household interview data, whereas our analysis used Medicaid claims data [28,29]. Furthermore, the CDC noted that 39% of the vaccinated adults who are 18 years or older received their influenza vaccine at a pharmacy or a store, 34% at a doctor’s office or health maintenance organization, and 8% at their workplace [29]. Our analyses focused on claims associated with a healthcare encounter. Therefore, we excluded vaccination administered at public facilities (e.g., senior/recreation/community centers), outpatient pharmacies, health departments, or workplaces [30], which may likely account for this difference.

Similar to the CDC and other studies, we observed that young children aged from 6 months to 4 years had the highest influenza vaccination uptake, likely due to well-child visits and vaccination schedule for other vaccines, and the 18–64 age group had the lowest vaccination uptake [29,30]. When comparing Medicaid’s national averages for influenza vaccination rate for 18–64 years, this study’s estimates were lower for every season considered; however, the trend matches the increasing vaccination rate until 2019, with a slight decrease in annual rates for Medicaid in 2020 [31]. Many reports also show that routine vaccination coverage for the Medicaid population is lower than those with commercial or private insurance coverage [32]. For example, data from the National Committee for Quality Assurance as well as Medicaid and CHIP Payment and Access Commission show flu vaccination for adults with Medicaid has significantly lower rates than adults on private health plans [7,31].

Factors associated with higher vaccine uptake during the 2016–2021 study period include sex, Medicaid managed care program, and living in a metropolitan area. Other studies have also noted higher rates of influenza vaccination in females, which persist even after controlling for important confounders [33,34]. We found that females have more frequent healthcare encounters, which likely results in an increased opportunity for healthcare providers to recommend vaccination [35]. Self-reported studies also noted that children and adults from urban areas had higher vaccination rates than those in rural areas [36,37]. Although Medicaid managed care contracts vary across states, having a higher vaccination rate among those who live in a state with a managed care system may be related to the fact that influenza is one of the immunizations identified as an adult health care quality measure in the Center for Medicare and Medicaid Services (CMS) adult core set measures [38]. In contrast to other studies that showed higher education associated with seasonal influenza vaccines, this study did not show such an association during the study period [39]. However, it is worth noting that the education data utilized in this study was derived from individuals residing in areas with education levels that were either higher or lower than the median, rendering it inequitable to compare with data acquired from individual self-reports or alternative definitions.

In this study, the vaccination rate was higher during peri-COVID compared to pre-COVID for all demographic categories considered, indicating a promising trend. However, it was slightly lower for high-risk groups, most likely due to the impact of COVID-19 on these groups and local travel restrictions. Increased vaccination rates in peri-COVID were consistent with a CDC report that showed a 9% increase in influenza vaccination rates during September–December 2020 compared to the same period in 2018 and 2019; however, the number of administrative doses declined among children during this time [40]. Similarly, another CDC report showed that during 2020–2021, influenza vaccine coverage was lower than during the 2019–2020 season for children but was reported to be higher for adults [29]. However, a recent study suggests that overall childhood vaccination rates have drastically decreased peri-COVID [41]. Understanding influenza vaccination rates, especially in vulnerable populations, will be important for population health policies moving forward.

We observed that Medicaid enrollees who were pregnant had higher rates of vaccine uptake compared to the general Medicaid population during the overall study period which is consistent with another Medicaid study reporting influenza vaccination during pregnancies (18%) [42]. Although authors did not find studies or reports that previously reviewed influenza vaccine uptake in the Medicaid diabetes population, our findings show that vaccination in individuals with diabetes is higher than in the general population which is consistent with other pre-COVID studies that looked at other populations with diabetes [43,44]. When comparing the vaccination rates for individuals who are pregnant or have diabetes during the peri-COVID era, we noted a decrease in vaccination rates compared to pre-COVID, a trend that is similarly seen in the overall study population. This decreased rate of vaccination during the peri-COVID era may in part be due to at-risk enrollees attempting to protect themselves and reduce exposure to the virus by reducing in-person healthcare encounters and clinic appointments that could be an opportunity for receiving the influenza vaccine [45]. Furthermore, we noticed that at-risk individuals who were vaccinated for influenza had more in-person healthcare encounters than those who were unvaccinated. These results are consistent with the findings that information about the vaccine effect for influenza during an in-person healthcare encounter predicts higher acceptance and uptake of influenza vaccination, especially in at-risk populations [43,46].

Our results further suggest an association between visits for general examination, evaluation, health maintenance encounters, or general mental health encounters and influenza vaccination during the study period. Pregnancy and related encounters, e.g., antenatal screening, liveborn delivery, or labor complications, were also strongly associated with increased vaccination rates for influenza. However, mental health (e.g., bipolar disorder or schizophrenia) and substance- or opioid-use disorders were among the predictors associated with low vaccine uptake. This observation is consistent with previous studies that have shown that persons with mental health, psychiatric, or substance use disorders are less likely to receive routine preventive healthcare services such as influenza vaccinations [47,48,49,50]. Some reasons reflected in the literature for a lower rate of vaccination in the population with mental health disorders are negative beliefs about the safety and efficacy of the vaccine, misperceptions that the influenza vaccine causes influenza, and perceiving that the influenza vaccine is not effective in protecting against influenza [50]. There is evidence that shows that patients with mental health disorders are at risk of physical comorbidities such as diabetes and cardiovascular conditions, which are, in turn, associated with a higher risk of influenza [50,51]. Thus, healthcare or mental health clinic settings may provide opportunities for communication of clinical education regarding the risks and benefits of influenza vaccination, especially for people who are at a higher risk of severe influenza outcomes.

This study also offers valuable opportunities to inform policymakers and enhance vaccination coverage across the Medicaid population. By identifying areas where influenza vaccine uptake falls below the desired benchmarks, this work equips policymakers with valuable data to develop targeted strategies for improving vaccination rates. The evidence underscores the critical importance of evaluating associations with vaccination to understand the determinants of low coverage rates, address disparities, and develop strategies to improve vaccination coverage [52]. These strategies can encompass tailored education and outreach programs designed to effectively reach at-risk groups and communicate the benefits of influenza vaccination. As part of a broader effort to address disparities in vaccine uptake, the findings of this study underscore the importance of proactive policy measures aimed at increasing population coverage for not only influenza but also other vaccine-preventable diseases. Ultimately, this work contributes to the evidence-based approach necessary for informed policy decisions and the implementation of measures to ensure the health and well-being of vulnerable populations.

## 5. Limitation

There are limitations to this study that impact the interpretation of the results. First, we only had access to claims data from nine states, of which six were only partial sets of claims. As a result, the findings cannot be generalizable across the whole U.S. Medicaid population despite the size of our study sample. In this study, we only included patients with a healthcare encounter to ensure patients had an opportunity to receive the influenza vaccine. Therefore, we do not include vaccines from pharmacies and other non-healthcare provider institutions. Another limitation is that the final quarter of the pre-COVID era would have been influenced by COVID-19, but we did not account for it in this study because a seasonal analysis was performed. Lastly, some of the CCS features associated with influenza vaccination are extremely broad, which can limit the interpretation of individual CCS features. It is important to note that the true vaccination status of an enrollee is unknown and strictly based on Medicaid claims data used in this study, which may bias the results. Additionally, although our data source is restricted to Medicaid claims, it is important to emphasize that Medicaid encompasses all vaccinations administered by pharmacies, employees, and vaccination campaigns. Nevertheless, a small number of vaccinations may be missed due to incomplete reporting or errors, as well as those that were not considered in this study, such as vaccinations administered at public facilities.

## 6. Conclusions

This study presents novel findings regarding influenza vaccine uptake among Medicaid enrollees and highlights opportunities to increase vaccine uptake among vulnerable populations as well as provide evidence that can be used by policymakers for education and outreach programs. Although at-risk groups (e.g., diabetes and pregnancy) have higher vaccination rates, it is still lower than the Healthy People 2030 population baseline and target range of 49.2–70.0% and justifies additional efforts to further improve influenza vaccine uptake. Although many factors may play a role in the lower rates of influenza vaccination observed in the Medicaid population, outreach and education may be needed to communicate the benefits of influenza vaccination (e.g., improving severe outcomes) and to improve these rates. In conclusion, learning the characteristics of unvaccinated groups can provide insight to help address the disparities in vaccine uptake for vaccine-preventable diseases, including COVID-19.

## Figures and Tables

**Figure 1 vaccines-11-01712-f001:**
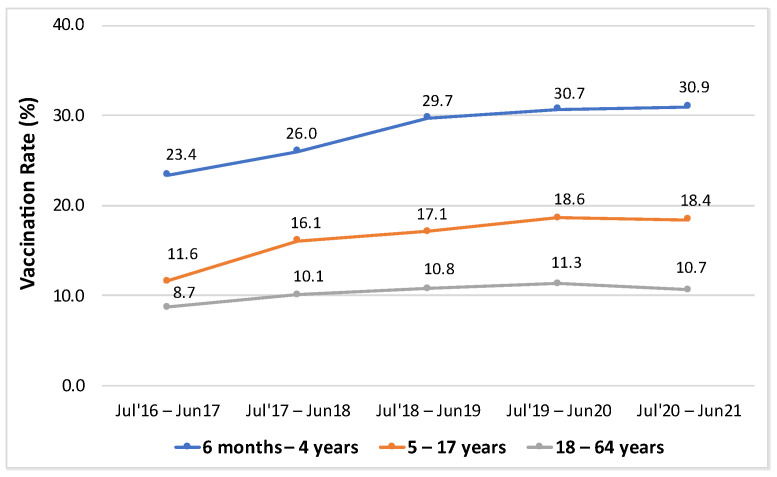
Vaccination rates stratified by age groups.

**Figure 2 vaccines-11-01712-f002:**
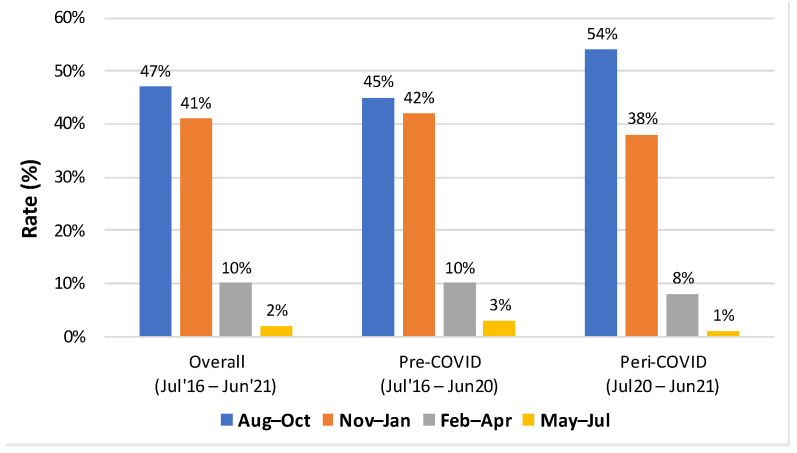
Sub-season vaccination rates among individuals who were vaccinated during the overall and pre- and peri-COVID era.

**Figure 3 vaccines-11-01712-f003:**
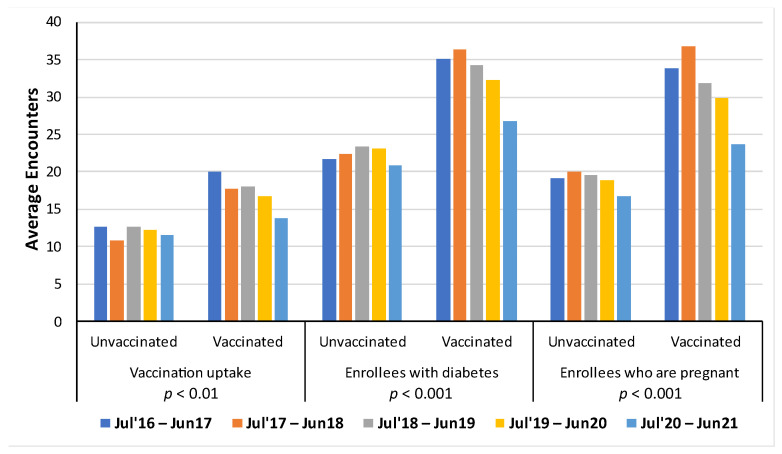
Average number of in-person healthcare encounters per influenza season.

**Table 1 vaccines-11-01712-t001:** Vaccination rates (%) based on demographics of enrollees with 1+ healthcare encounters, stratified by COVID-19 Time Periods.

	Overall (July 2016–June 2021)			Pre-COVID (July 2016–June 2020)			Peri-COVID (July 2020–June 2021)		
	Vaccinated	Total Enrollees	%	Vaccinated	Total Enrollees	%	Vaccinated	Total Enrollees	%
Total	3,050,471	20,868,910	15	2,407,644	16,807,523	14	642,827	4,061,387	16
Age Groups									
6 months–4 years	734,672	2,565,290	29	562,144	2,007,435	28	172,528	557,855	31
5 years–17 years	1,113,634	6,706,443	17	883,619	5,457,074	16	230,015	1,249,369	18
18 years–64 years	1,202,165	11,597,177	10	961,881	9,343,014	10	240,284	2,254,163	11
Gender									
Female	1,714,621	11,596,733	15	1,353,614	9,308,460	15	361,007	2,288,273	16
Male	1,335,850	9,272,177	14	1,054,030	7,499,063	14	281,820	1,773,114	16
Metropolitan Area									
Metropolitan	2,452,070	16,686,225	15	1,924,884	13,357,713	14	527,186	3,328,512	16
Non-Metropolitan	598,401	4,182,685	14	482,760	3,449,810	14	115,641	732,875	16
Education									
Greater than Median Higher Education Area	1,380,567	9,410,850	15	1,088,778	7,567,529	14	291,789	1,843,321	16
Lower than Median Higher Education Area	1,669,904	11,458,060	15	1,318,866	9,239,994	14	351,038	2,218,066	16
Medicaid Managed Care									
With Managed Care Program	1,249,118	7,537,820	17	985,363	5,978,761	16	263,755	1,559,059	17
Free-for-service	1,801,353	13,331,090	14	1,422,281	10,828,762	13	379,072	2,502,328	15
At-risk Health Conditions									
Diabetes *	554,671	3,348,282	17	462,789	2,755,810	17	91,882	592,472	16
Diabetes Type 1 (DM 1)	103,442	542,915	19	87,351	448,292	19	16,091	94,623	17
Diabetes Type 2 (DM 2)	530,056	3,174,874	17	442,665	2,614,529	17	87,391	560,345	16
Pregnancy	593,094	3,423,662	17	442,533	2,558,276	17	33,184	203,434	16

* Enrollees in this category were diagnosed as either: (1) “DM 1 and DM 2”, (2) “DM 1 or DM2”, or (3) “unspecified diabetes”.

**Table 2 vaccines-11-01712-t002:** Top and bottom 10 adjusted odds ratios for CCSR features associated with influenza vaccination uptake among enrollees with at least one healthcare encounter stratified by COVID era.

	Overall (July 2016–June 2021)	Pre-COVID (July 2016–June 2020)	Peri COVID (July 2020–June 2021)
Demographics Characteristics			
6 months–4 years (vs. 18–64 years)	3.11 (3.09, 3.14)	3.25 (3.22, 3.27)	2.88 (2.85, 2.92)
5–17 years (vs. 18–64 years)	1.91 (1.9, 1.92)	1.93 (1.92, 1.94)	1.90 (1.88, 1.91)
Male (vs. Female)	0.97 (0.97, 0.97)	0.97 (0.97, 0.98)	0.95 (0.94, 0.96)
Area with Higher than Median Education (vs. Not)	1.00 (1, 1.01)	1.01 (1, 1.01)	1.00 (0.99, 1)
Metropolitan (vs. Non-Metropolitan)	1.01 (1.01, 1.02)	0.99 (0.98, 0.99)	1.00 (1, 1.01)
Managed Care (vs. Non-Managed Care)	1.18 (1.17, 1.18)	1.21 (1.21, 1.22)	1.10 (1.1, 1.11)
CCSR Diabetes Mapped Categories			
Diabetes mellitus with complication	1.06 (1.05–1.07)	1.07 (1.07–1.08)	1.04 (1.03–1.05)
Diabetes mellitus without complication	1.01 (1.01–1.02)	1.01 (1.00–1.01)	1.02 (1.01–1.03)
CCSR Pregnancy Mapped Categories			
Antenatal screening	1.34 (1.33–1.36)	1.29 (1.27–1.30)	1.17 (1.15–1.19)
Malposition, disproportion, or other labor complications	1.22 (1.20–1.23)	1.18 (1.16–1.19)	1.18 (1.16–1.21)
Supervision of high-risk pregnancy	1.13 (1.19–1.14)	1.12 (1.11–1.14)	1.08 (1.06–1.10)
Other specified complications in pregnancy	1.13 (1.11–1.14)	1.10 (1.09–1.16)	1.13 (1.11–1.15)
Maternal outcome of delivery	1.12 (1.10–1.13)	1.10 (1.08–1.11)	1.01 (0.99–1.03)
Maternal care related to fetal conditions	1.10 (1.09–1.12)	1.08 (1.07–1.10)	1.05 (1.03–1.07)
Uncomplicated pregnancy, delivery, or puerperium	1.08 (1.07–1.09)	1.09 (1.08–1.10)	0.97 (0.95–0.99)
Gestational weeks	0.77 (0.76–0.78)	0.80 (0.79–0.81)	0.76 (0.75–0.77)
Top 10 CCSR Condition Categories			
Contact with Health Services (Medical examination/evaluation)	2.07 (2.06, 2.08)	1.91 (1.9, 1.92)	1.91 (1.89, 1.92)
Encounter for Perinatal Period (Liveborn)	1.79 (1.77, 1.8)	1.41 (1.4, 1.42)	1.59 (1.57, 1.6)
Encounter for Pregnancy, Childbirth, and the Puerperium (Antenatal screening)	1.34 (1.33, 1.36)	1.29 (1.27, 1.3)	1.17 (1.15, 1.19)
Contact with Health Services (Encounter for observation and examination, excludes infectious disease, neoplasm, mental disorders)	1.29 (1.28, 1.29)	1.24 (1.24, 1.25)	1.22 (1.21, 1.22)
Contact with Health Services (Neoplasm-related encounters)	1.27 (1.26, 1.27)	1.25 (1.25, 1.26)	1.25 (1.24, 1.26)
Contact with Health Services (Mental health conditions)	1.26 (1.24, 1.27)	1.10 (1.09, 1.11)	1.41 (1.39, 1.42)
Encounter for Pregnancy, Childbirth, and the Puerperium (Labor complications)	1.22 (1.2, 1.23)	1.18 (1.16, 1.19)	1.18 (1.16, 1.21)
Encounter for Eye and Adnexa (Refractive error)	1.22 (1.21, 1.22)	1.21 (1.21, 1.22)	1.05 (1.04, 1.06)
Encounter for Eye and Adnexa (Cataract and other lens disorders)	1.21 (1.2, 1.22)	1.24 (1.22, 1.25)	1.13 (1.11, 1.15)
Encounter for Respiratory system (Other specified upper respiratory infections)	1.21 (1.2, 1.21)	1.22 (1.21, 1.22)	1.03 (1.02, 1.04)
Bottom 10 CCSR Condition Categories			
Encounter for Circulatory System (Heart failure)	0.90 (0.89, 0.91)	0.88 (0.87, 0.89)	0.92 (0.91, 0.94)
Encounter for Diseases of the Genitourinary System (Menopausal disorders)	0.89 (0.88, 0.9)	0.91 (0.9, 0.92)	0.89 (0.87, 0.9)
Encounter for Nervous System Disorder (Epilepsy; convulsions)	0.89 (0.88, 0.9)	0.90 (0.9, 0.91)	0.97 (0.95, 0.99)
Encounter for Mental, Behavioral, and Neurodevelopmental Disorders (Tobacco-related disorders)	0.89 (0.88, 0.9)	0.90 (0.9, 0.91)	0.89 (0.87, 0.9)
Encounter for Mental, Behavioral, and Neurodevelopmental Disorders (Bipolar and related disorders)	0.87 (0.87, 0.88)	0.88 (0.87, 0.89)	0.85 (0.84, 0.86)
Encounter for Mental, Behavioral, and Neurodevelopmental Disorders (Schizophrenia spectrum and other psychotic disorders)	0.87 (0.86, 0.88)	0.89 (0.88, 0.9)	0.90 (0.89, 0.91)
Encounter for Mental, Behavioral, and Neurodevelopmental Disorders (Opioid-related disorders)	0.84 (0.83, 0.85)	0.87 (0.86, 0.88)	0.84 (0.83, 0.85)
Encounter for Diseases of the Genitourinary System (Inflammatory diseases of female pelvic organs)	0.83 (0.83, 0.84)	0.85 (0.85, 0.86)	0.76 (0.75, 0.77)
Encounter for Pregnancy, Childbirth, and the Puerperium (Gestational weeks)	0.83 (0.82, 0.84)	0.85 (0.84, 0.85)	0.84 (0.83, 0.85)
Contact with Health Services (No immunization or under immunization)	0.62 (0.61, 0.62)	0.65 (0.64, 0.65)	0.51 (0.5, 0.52)

## Data Availability

Data sharing is not applicable to this article.

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
