# Peer review of "Trends in Influenza Vaccination Rates among a Medicaid Population from 2016 to 2021"

_vaccines, 2023, doi:10.3390/vaccines11111712_

Round 1

Reviewer 1 Report

Comments and Suggestions for Authors

Dear authors

This study provides information on the status of influenza vaccination in an area.

The writing is good and study design and population is ok. 

The study needs adding insights for policy-making and population coverage of vaccination. 

The reason for differences among gender or age groups in terms of vaccination rate can be added. 

Comments on the Quality of English Language

Dear editor

The English writing of the study is suitable. 

Author Response

Response to Reviewers

Reviewer #1

This study provides information on the status of influenza vaccination in an area. The writing is good and study design and population is ok.

The study needs adding insights for policy-making and population coverage of vaccination.

Reply: We thank the Reviewer for the opportunity to improve the study content. We have added the following paragraph to the discussion section:

This study offers valuable opportunities to inform policymakers and enhance vaccination coverage across the Medicaid population. By identifying areas where influenza vaccine uptake falls below the desired benchmarks, this work equips policymakers with valuable data to develop targeted strategies for improving vaccination rates. Evidence underscores the critical importance of evaluating associations with vaccination to understand the determinants of low coverage rates, address disparities, and develop strategies to improve vaccination coverage (Williams et al., 2016; Centers for Disease Control and Prevention, 2016). These strategies can encompass tailored education and outreach programs designed to effectively reach at-risk groups and communicate the benefits of influenza vaccination. As part of a broader effort to address disparities in vaccine uptake, the findings of this study underscore the importance of proactive policy measures aimed at increasing population coverage for not only influenza but also other vaccine-preventable diseases. Ultimately, this work contributes to the evidence-based approach necessary for informed policy decisions and the implementation of measures to ensure the health and well-being of vulnerable populations.

The reason for differences among gender or age groups in terms of vaccination rate can be added.

Reply: The differences by gender and age are important and consistent with existing literature. We discuss this in detail on Page 22 in the discussion, including a discussion on child-well visit examinations and the likelihood this contributes to higher vaccination

Reviewer 2 Report

Comments and Suggestions for Authors

Estimated Authors,

I've been invited to review the present paper, entitled "TRENDS IN INFLUENZA VACCINATION RATES AMONG A MEDICAID POPULATION, 2016-2021".

Through a retrospective analysis of MEDICAID data, authors examined the trend 2016 to 2021 of influenza vaccination, and ascertained the main individual factors associated with vaccination status, both positively (facilitators) and negatively (barriers).

The main significance of this study is due to the high number of enclosed individuals, while the main barriers are due to the very same quasi-experimental study design, with retrospective data retrieved from a large institutional database.

From my point of view, the present study could be accepted for publication but several improvements are required. Most notably:

1) the results section is quite confusing, as it is often unclear when Authors are reporting data on the overall study population or on the vaccinated population. Please double check and revise the flow of the main text in order to avoid any potential confusion;

2) limits section should acknowledge the design limits, associated with having retrieved the whole of included data from an institutional database, being therefore unable to ascertain the individual personal and behavioural factors;

3) Authors should explain to the international readers which patients are actually assisted by Medicaid, whether the inclusion criteria have been modified across the study period or not, and include more extensively this potential shortcoming in the limits secation

  Comments on the Quality of English Language

Appropriate quality but somehow confusing in the results section, that could be fixed.

Author Response

Response to Reviewers

Reviewer #2

I've been invited to review the present paper, entitled "TRENDS IN INFLUENZA VACCINATION RATES AMONG A MEDICAID POPULATION, 2016-2021". Through a retrospective analysis of MEDICAID data, authors examined the trend 2016 to 2021 of influenza vaccination, and ascertained the main individual factors associated with vaccination status, both positively (facilitators) and negatively (barriers). The main significance of this study is due to the high number of enclosed individuals, while the main barriers are due to the very same quasi-experimental study design, with retrospective data retrieved from a large institutional database. From my point of view, the present study could be accepted for publication but several improvements are required. Most notably:

1) the results section is quite confusing, as it is often unclear when Authors are reporting data on the overall study population or on the vaccinated population. Please double check and revise the flow of the main text in order to avoid any potential confusion;

Reply: We thanks the Reviewer for the opportunity to improve the manuscript. In the revised manuscript, we addressed this issue. We have significantly reduced the results section (as suggested by another Reviewer) and improved the flow of the results section.

In response to the reviewer’s question, we considered all individuals with valid Medicaid enrollment and at least one encounter in each vaccination season and calculated vaccination rates for each season based on those with at least one influenza vaccination claim using the codes shown in Table S4.

2) limits section should acknowledge the design limits, associated with having retrieved the whole of included data from an institutional database, being therefore unable to ascertain the individual personal and behavioural factors;

Reply: We have clarified our data source throughout the manuscript. The data are derived from State-wide Medicaid claims, which would not be consider as an “institutional database”. According to FDA guidance, these types of data are commonly referred to as real-world data (RWD) sources. https://www.fda.gov/science-research/science-and-research-special-topics/real-world-evidence

3) Authors should explain to the international readers which patients are actually assisted by Medicaid, whether the inclusion criteria have been modified across the study period or not, and include more extensively this potential shortcoming in the limits section.

Reply: We included the following paragraph in the introduction section:

“Medicaid, a federally funded healthcare program in the United States provides health coverage to eligible low-income adults, children, pregnant people, elderly adults and people with disabilities (Medicaid.gov, 2023b). Additionally, Medicaid serves other vulnerable groups, such as foster children and refugees, offering them essential access to healthcare services.”

Comments on the Quality of English Language

Appropriate quality but somehow confusing in the results section, that could be fixed.

Reviewer 3 Report

Comments and Suggestions for Authors

I was invited to revise the paper entitled "TRENDS IN INFLUENZA VACCINATION RATES AMONG A MEDICAID POPULATION, 2016-2021". It was a cohort study aimed to evaluate the influenza vaccination rates among Medicaid enrollees using Medicaid claims data. 

The study focused on a important topic and analyzed a large datased reporting standardized data. 

The methodology proposed was appropriate and strong.

Observations:

- I suggest to evaluate also annual trend with appropriate statistical models (jointpoint regression?);

- It is unclear if in table 1 authors reported the number of patients totally vaccinated or the number of doses performed. THe aggregated data in unclear and hard to be interpreted as reported in table 1;

- Despite the novelty of this analysis regarding DM and pregnancy, also other conditions have to be considered as covariates;

- Why did Authors divided age categories into these three groups? The distribution of patients was unequal among groups and it can led to misinterpretation of data.

- THe analysis reported in Figure 3 need to report the relativ p-value;

- Top 10 CCSR reported in table 2 should also be added in table 1;

- In table 2 should be reported as appendix the covariates used in the relative model;

- In introduction section Authors should report how flu vaccination campaign was scheduled and organized. In particular Authors should describe all categories of patients enrolled in the flu vaccination campaign.

Author Response

Response to Reviewers

Review #3

I was invited to revise the paper entitled "TRENDS IN INFLUENZA VACCINATION RATES AMONG A MEDICAID POPULATION, 2016-2021". It was a cohort study aimed to evaluate the influenza vaccination rates among Medicaid enrollees using Medicaid claims data. The study focused on a important topic and analyzed a large datased reporting standardized data. The methodology proposed was appropriate and strong.

Observations:

- I suggest to evaluate also annual trend with appropriate statistical models (jointpoint regression?);

Reply: Joinpoint regression is a valuable approach, and I appreciate the reviewer for recommending it.   Nevertheless, the duration of our study (2016–2021) is relatively short, making the use of joinpoint regression less advisable for the yearly vaccination rate. The statistical technique of segmented regression is employed to analyze long-term time series data in cases where there is suspicion of multiple segments with distinct linear trends or growth rates in the underlying data. This is included in the results section.

- It is unclear if in table 1 authors reported the number of patients totally vaccinated or the number of doses performed. THe aggregated data in unclear and hard to be interpreted as reported in table 1;

Reply: We only considered individuals who have Medicaid coverage during each vaccine season and who have had at least one encounter. This is because the eligibility of enrollees for the Medicaid program may vary over time. For each season, we evaluated the eligibility of individuals and their vaccinations independently and calculated the total number by adding the seasons together for the given era. Table 1 displays the total number of enrollees who have enrolled in or have valid Medicaid coverage and have had at least one encounter during the vaccine season. Out of these enrollees, those who have made a single claim for the influenza vaccine are classified as vaccinated enrollees. Table 1 presents the two statistics (total enrollees and vaccinated enrollees) and vaccine rate for three time periods. To clarify this issue, we added the following sentence to the second paragraph of the results section:

“Table 1 shows the vaccination rates during the pre-COVID and peri-COVID eras. Total enrollees in this table indicate the total number of individuals who were enrolled in or had valid Medicaid coverage in the given COVID-19 era and had at least one healthcare encounter during each influenza season. Enrollees were counted separately per season and summed up to calculate the total number for the given COVID-19 era. The total vaccinated enrollees are the total of influenza vaccine claims made during the influenza seasons of the given COVID-19 era. Only one influenza vaccine claim was counted per enrollee during each influenza season.”

- Despite the novelty of this analysis regarding DM and pregnancy, also other conditions have to be considered as covariates;

Reply: One of our hypotheses was evaluating vaccination rates on vulnerable enrollees, including diabetes and pregnancy, and we have highlighted this in the manuscript. In addition, we evaluated other CCSR conditions as covariates in the study and reported the top 10 covariates in Table 2 and all CCSR covariates in Table S7.

- Why did Authors divided age categories into these three groups? The distribution of patients was unequal among groups and it can led to misinterpretation of data.

Reply: The age groups were determined based on CDC (Centers for Disease Control and Prevention) age categorization, which can be found at https://www.cdc.gov/flu/fluvaxview/trends/age-groups.htm. The only difference is that the group 6 months–17 years was divided into two subgroups, 6 months–4 and 5–17, because the CDC highlights that those under 5 are more likely to develop serious complications (https://www.cdc.gov/flu/professionals/vaccination/vax-summary.htm)

This was previously stated in the following manner in the manuscript: “These age groups were developed based on the CDC recommendations for persons aged 6 months and older to receive an annual influenza vaccine”. We added the aforementioned link to our reference section. 

- THe analysis reported in Figure 3 need to report the relativ p-value;

Reply: We added p values to Figure 3 for each group: Vaccination uptake (p<0.01), Enrollees with diabetes (p<0.001) and Enrollees who are pregnant (p<0.001).

- Top 10 CCSR reported in table 2 should also be added in table 1;

Reply: Table 1 presents population-level characteristics that are crucial for policymakers in formulating future vaccination plans. Conversely, Table 2 indicates the individual-level factors that predict vaccine uptake, obtained through a logistic regression model. As the objectives of the two tables are entirely distinct, we did not want to confuse the reader by including CCSR features.

- In table 2 should be reported as appendix the covariates used in the relative model;

Reply: We added all CCSR covariates used in the model in an additional supplementary table (Table S7) with both odds ratios and vaccination rates.

- In introduction section Authors should report how flu vaccination campaign was scheduled and organized. In particular Authors should describe all categories of patients enrolled in the flu vaccination campaign.

Reply: This work is not part of a flu vaccination campaign, but rather reports flu vaccination in the federal Medicaid program. To clarify this, we have added a paragraph to the introduction section:

“Medicaid, a federally funded healthcare program in the United States provides health coverage to eligible low-income adults, children, pregnant people, elderly adults and people with disabilities (Medicaid.gov, 2023b). Additionally, Medicaid serves other vulnerable groups, such as foster children and refugees, offering them essential access to healthcare services.”

Reviewer 4 Report

Comments and Suggestions for Authors

TRENDS IN INFLUENZA VACCINATION RATES AMONG A MEDICAID POPULATION, 2016-2021

I think this study is a useful contribution to the field of influenza research and provides helpful data on trends in vaccination data.

Summary: This study evaluated trends in vaccination rates of Medicaid enrollees from 6 months to 65 years through Medicaid claims data. The study broke down certain time periods around the COVID-19 pandemic.

Introduction:

·       P1 – Why cutoff incidence rate at below 65? At least for introduction I would present rates for all age ranges especially since vaccination is important for the senior population and I assume the other stats in paragraph 1 include that age group. Then I see in methods you explain that only Medicaid claims were used limiting to 64 and below. Was there a possibility to include Medicare data as well?

·       P4 – Most of this paragraph describing the data structure belongs in the methods section.

·       P5 – Why are there limited years of available data?

Materials and Methods:

·       Data Source – How were the states chosen? Can you list them or describe them? Are they representative of the US population (i.e. rurality, political beliefs, industries, economic prospects or other social support, changes to childhood vaccine laws – factors that might influence vaccination rates)?

·       Measures – The May-July sub-season straddles two influenza seasons July-June. Why the discrepancy?

·       Measures – The last quarter of the pre-Covid year would have been impacted by COVID-19 lock downs. Additionally, COVID-19 in China earlier in the year may have impacted vaccination rates earlier in the influenza season. Did you look at 2020 in isolation, or pull it out of the pre-covid data to see if the rates differed from other pre-covid years and the peri-Covid year?

·       Measures – How many CCSR categories are there?

·       Outcomes – I believe the table reference should be changed from S5 to S4.

·       Statistical Analysis – Describe how you defined/calculated vaccination rate – ex) At least one CPT code identified per enrollee in an influenza season divided by the total number of enrollees for that season.

·       Statistical Analysis – I am a bit confused what your primary exposure of interest is here. It seems like you are just looking at trends, however, in the introduction you make a hypothesis about people with diabetes and people who are pregnant. You define those groups in S1 and S2 however do not include them in your logistic regression model. If you are using the CCSR to identify them, then you should define those categories based on how the CCSR does or affirm that table S1 and S2 follow CCSR designations.

·       Statistical Analysis – State that you will look at the highest 10 and lowest 10 odds ratios for CCSR categories.

·       Statistical Analysis - Fisher’s exact test was used to calculate the unadjusted odds ratio to explore independent associations of covariates with flu vaccination – this is only used for small datasets I am unsure what this was used for in such a large dataset and I also don’t see these in the results section anywhere

Results:

·       Vaccination Rates during Pre-COVID and Peri-COVID era P3 – the () are flipped compared to the previous paragraphs making it challenging to read, suggest switching () from the percent to the N= results.

·       Table 2 – Since pregnancy and diabetes were part of your hypothesis the Odds ratios as identified in tables S1 and S2 should be presented in this table

·       From the methods - We used Chi-square tests of independence to assess the difference in vaccination rates across different flu seasons.- Am I missing something or were these results not shared in the results section. I just see the table in the supplemental materials but it wasn’t referenced or discussed in the results.

·       General comment – the text repeats each table and figure verbatim, the text can be summarized quite a bit and reference the tables for more detail – the word count can be cut significantly in this section.

Discussion

·       P2- Since this study did not account for community based vaccination such as pharmacy/employer/community centers, the wording should state “Rates of influenza vaccination identified through Medicaid claims data were lower” instead of “In this Medicaid population, rates of influenza vaccination were lower”, it is misleading to say this Medicaid population since your data source was limited to claims and is not representative over overall vaccination rates since you don’t know the true vaccination rate in the Medicaid population.

o   I wonder if the introduction should be more explicit in the research question of interest. Vaccination rates among those that got vaccinated at a medical facility accepting Medicaid.

·       “In contrast to other studies that showed higher education associated with seasonal influenza vaccines, this study did not show such association during the study period (Li et al., 2022).” – Li defined education differently than you, this should be stated here.

·       “Our results further suggest that visits for general examination, evaluation, health maintenance encounters, or general mental health encounter is a strong predictor of influenza vaccination for the study period.” – I don’t think your data backs up this statement – did you look at temporality with vaccination? It can only be a predictor if it happened before the vaccine. Maybe vaccination predicts number of visits?

Author Response

Response to Reviewers

Reviewer #4

I think this study is a useful contribution to the field of influenza research and provides helpful data on trends in vaccination data.

Summary: This study evaluated trends in vaccination rates of Medicaid enrollees from 6 months to 65 years through Medicaid claims data. The study broke down certain time periods around the COVID-19 pandemic.

Introduction:

  • P1 – Why cutoff incidence rate at below 65? At least for introduction I would present rates for all age ranges especially since vaccination is important for the senior population and I assume the other stats in paragraph 1 include that age group. Then I see in methods you explain that only Medicaid claims were used limiting to 64 and below. Was there a possibility to include Medicare data as well?

Reply: Our access was limited to only Medicaid data, so we selected the age cutoff at 65 because the Medicare program specifically provides coverage for individuals who are 65 years or older. Thus, including individuals over the age of 65 may encompass those who are eligible for both Medicaid and Medicare, leading to an inaccurate vaccination rate. This was previously discussed in the second paragraph of Section Measures in the manuscript: “Individuals who were aged 65 years and older were not included in this study as they are likely to be dually enrolled in Medicare, which would render their Medicaid claims an incomplete record of their total utilization.”

  • P4 – Most of this paragraph describing the data structure belongs in the methods section.

Reply: In this paragraph, we tried to provide a brief introduction of the Medicaid administrative claims data that is our source data and its importance. We have moved some of this to the methods section, as suggested by the Reviewer.

  • P5 – Why are there limited years of available data?

Reply: As mentioned in the methods section, data were only available from 2016-2021. Although this is a limited number of years, this work represents one of the most comprehensive studies on flu vaccinations in this population.

Materials and Methods:

  • Data Source – How were the states chosen? Can you list them or describe them? Are they representative of the US population (i.e. rurality, political beliefs, industries, economic prospects or other social support, changes to childhood vaccine laws – factors that might influence vaccination rates)?

Reply: This study consisted of de-identified enrollee data obtained from Medicaid enrollment and claims provided by the Digital Health Cooperative Research Center. Data collected were from three western states, two midwestern states and two southeastern states between January 2016 and June 2020. Two states provide data on the entire Medicaid population including both the managed care plan and the fee-for-service patients. Four states provide data from a managed care plan and one state had only fee-for-service Medicaid beneficiaries. Due to data-use-agreement contract, we are not allowed to identify the exact states. We have added this information to the methods section.

  • Measures – The May-July sub-season straddles two influenza seasons July-June. Why the discrepancy?

Reply: The vaccination rates during the initial and final months of each season (July and June) were extremely low. We utilized these two months, along with May, in our final subgroup to better illustrate the segmentation of vaccine uptake across sub-seasons. Figure S1 illustrates sub-season vaccination rates across all seasons, showing significant differences between sub-season vaccination rates (p < 0.05). We added this clarification to the method section as well as the result section, where sub-season vaccination rates are reported.

During the 201x season, which spanned from July 201x to June 201x+1, the last subgroup included July 201x, May 201x+1, and June 201x+1.

  • Measures – The last quarter of the pre-Covid year would have been impacted by COVID-19 lock downs. Additionally, COVID-19 in China earlier in the year may have impacted vaccination rates earlier in the influenza season. Did you look at 2020 in isolation, or pull it out of the pre-covid data to see if the rates differed from other pre-covid years and the peri-Covid year?

Reply: We did not look at 2020 in isolation, but the Reviewer is correct, and we addressed this issue in the limitation section as follows:

“Another limitation is that the final quarter of the pre-Covid era would have been influenced by COVID-19, but we did not account for it in this study because a seasonal analysis was performed.”

  • Measures – How many CCSR categories are there?

Reply: All conditions were mapped to 166 CCSR categories. We clarified this in the result section in the third paragraph of Subsection “Variables Associated with Influenza Vaccination Uptake” as follows:

“All conditions associated with enrollees were mapped to 166 CCSR categories in the logistic regression model. We investigated CCSR features that are most strongly associated with influenza vaccination uptake in the Medicaid population (Table 2).”

  • Outcomes – I believe the table reference should be changed from S5 to S4.

Reply: Thank you for identifying the error. We updated the table references accordingly.

  • Statistical Analysis – Describe how you defined/calculated vaccination rate – ex) At least one CPT code identified per enrollee in an influenza season divided by the total number of enrollees for that season.

Reply: This was addressed in response to Reviewer 3’s second comment, in which the elements of Table 1 were explained. We also described the calculation of the vaccination rate in the Measures section, as the reviewer suggested.

“To calculate the vaccination rate for each season, we divided the total number of vaccinated enrollees with at least one vaccination claim (as shown in Table S4) by the total number of enrollees who have enrolled in or have valid Medicaid coverage and at least one healthcare encounter during the season.”

To calculate the vaccination rate for an era, the total vaccinated enrollees and total enrollees were obtained by adding those values during the included seasons.” 

  • Statistical Analysis – I am a bit confused what your primary exposure of interest is here. It seems like you are just looking at trends, however, in the introduction you make a hypothesis about people with diabetes and people who are pregnant. You define those groups in S1 and S2 however do not include them in your logistic regression model. If you are using the CCSR to identify them, then you should define those categories based on how the CCSR does or affirm that table S1 and S2 follow CCSR designations.

Reply: We utilized a searchable list of ICD-10-CM diagnosis codes mapped to CCSR categories, including default assignment for principal diagnosis, v2021.2 at https://hcup-us.ahrq.gov/toolssoftware/ccsr/dxccsr.jsp. As a result, all codes from Tables S1 and S2 that were observed in the study cohort’s individuals were mapped to the corresponding CCSR categories and incorporated into the logistic regression model. However, their odds ratio did not place them among the top 10 features. Nevertheless, the purpose of the regression analysis was to identify the CCSR-based factors that were predictive of person-based vaccination. In fact, the observed vaccination trend among the vulnerable population may not necessarily be indicative of their predictive ability, and we did not intend for our regression analysis to imply this. Below is a sample of the mapping tables proposed by the CCSR software tool noted above:

ICD-10-CM Code

ICD-10-CM Code Description

CCSR Category

CCSR Category Description

Z3492

Encounter for supervision of normal pregnancy, unspecified, second trimester

PRG029

Uncomplicated pregnancy, delivery or puerperium

Z36

Encounter for antenatal screening of mother

PRG001

Antenatal screening

E1011

Type 1 diabetes mellitus with ketoacidosis with coma

END003

Diabetes mellitus with complication

We added the CCSR software tool link as a reference and clarified this in the Measures section as follows:

“We utilized a searchable list of ICD-10-CM diagnosis codes mapped to CCSR categories, including default assignment for principal diagnosis, v2021.2 (CCSR Software Tool, 2021). As a result, all codes found in the study cohort (including those in Tables S1 and S2) were mapped to the corresponding CCSR categories and incorporated into the logistic regression model.”

  • Statistical Analysis – State that you will look at the highest 10 and lowest 10 odds ratios for CCSR categories.

Reply: We revised the end of the section as follows:

“We utilized a searchable list of ICD-10-CM diagnosis codes mapped to CCSR categories, including default assignment for principal diagnosis, v2021.2 (CCSR Software Tool, 2021). As a result, all codes found in the study cohort (including those in Tables S1 and S2) were mapped to the corresponding CCSR categories and incorporated into the logistic regression model. An analysis was conducted to examine the top 10 and bottom 10 odds ratios for CCSR categories.”

  • Statistical Analysis - Fisher’s exact test was used to calculate the unadjusted odds ratio to explore independent associations of covariates with flu vaccination – this is only used for small datasets I am unsure what this was used for in such a large dataset and I also don’t see these in the results section anywhere

Reply: Since there was no notable disparity between the top and bottom CCSR features selected by both adjusted and unadjusted odds ratios, we only retained the adjusted ones. However, we inadvertently overlooked excluding them from the method section. We excluded it from the statistical analysis section of the revised manuscript.

Results:

  • Vaccination Rates during Pre-COVID and Peri-COVID era P3 – the () are flipped compared to the previous paragraphs making it challenging to read, suggest switching () from the percent to the N= results.

Reply: We changed them as follows and adhered to the format used in the preceding sections. We also summarized it based on the other reviewers’ comments.

“Enrollees living in a metropolitan area had a higher overall vaccination rate of 15% (N=2,452,070) compared to enrollees not living in a metropolitan area, 14% (N=598,401), and both had higher vaccination rates during the peri-COVID period compared to the pre-COVID period. A lower vaccination rate was observed for states that only had FFS during both pre-COVID and peri-COVID compared to Medicaid managed care states.”

  • Table 2 – Since pregnancy and diabetes were part of your hypothesis the Odds ratios as identified in tables S1 and S2 should be presented in this table

Reply: The CCSR categories linked to the codes in Tables S1 and S2, which were observed in the study cohort, were incorporated into Table 2 as a distinct group, as their odds ratios did not fall within the highest or lowest 10 features. The additional rows are as follows:

Overall                      (Jul'16-Jun'21)

Pre-COVID (Jul'16-Jun'20)

Peri COVID

(Jul'20-Jun'21)

CCSR Diabetes

Diabetes mellitus with complication

1.063 (1.055-1.072)

1.074 (1.066-1.083)

1.039 (1.026-1.052)

Diabetes mellitus without complication

1.014 (1.007-1.022)

1.006 (0.999-1.014)

1.017 (1.005-1.028)

CCSR Pregnancy

Antenatal screening

1.342 (1.327-1.358)

1.288 (1.273-1.304)

1.171 (1.149-1.193)

Malposition, disproportion or other labor complications

1.217 (1.201-1.233)

1.175 (1.159-1.191)

1.182 (1.157-1.207)

Supervision of high-risk pregnancy

1.131 (1.118-1.144)

1.121 (1.108-1.135)

1.081 (1.061-1.101)

Other specified complications in pregnancy

1.126 (1.114-1.139)

1.102 (1.090-1.115)

1.127 (1.107-1.148)

Maternal outcome of delivery

1.117 (1.104-1.131)

1.097 (1.083-1.112)

1.007 (0.986-1.027)

Maternal care related to fetal conditions

1.104 (1.091-1.117)

1.082 (1.068-1.095)

1.054 (1.033-1.074)

Uncomplicated pregnancy, delivery or puerperium

1.08 (1.069-1.091)

1.087 (1.076-1.098)

0.969 (0.953-0.985)

Gestational weeks

0.772 (0.764-0.781)

0.802 (0.793-0.811)

0.759 (0.745-0.773)

We also added the following statement in the result section:

“The adjusted odds ratios for two CCSR diabetic categories observed in the overall cohort study demonstrated a weak association between diabetes and vaccination uptake. In contrast, for most of the pregnancy CCSR categories, we observed a stronger association, indicating that pregnant enrollees were more likely to receive an influenza vaccination. Table S7 presents odds ratios for all CCSR features used in the logistic regression model, including patient number and vaccination rate.”

  • From the methods - We used Chi-square tests of independence to assess the difference in vaccination rates across different flu seasons.- Am I missing something or were these results not shared in the results section. I just see the table in the supplemental materials but it wasn’t referenced or discussed in the results.

Reply: The chi-square test was employed to assess whether there were statistically significant disparities in the vaccination rates between flu seasons, based on the group characteristics presented in Table S5.   We added the following information as an annotation in Table S5.

“P values were obtained from the chi-square test to show the significance of vaccination rates across influenza seasons.”

We also included it in the results section after Figure 1, vaccination rates stratified by age groups, in the following manner:

 “Table S5 shows the vaccination rates for all cohort characteristics throughout all influenza seasons. The differences between vaccination rates across influenza seasons were significant (p < 0.0001) using chi-square test.”

  • General comment – the text repeats each table and figure verbatim, the text can be summarized quite a bit and reference the tables for more detail – the word count can be cut significantly in this section.

Reply: We summarized this section by removing the repeated parts and trying to highlight specific aspects of the results.

Discussion

  • P2- Since this study did not account for community based vaccination such as pharmacy/employer/community centers, the wording should state “Rates of influenza vaccination identified through Medicaid claims data were lower” instead of “In this Medicaid population, rates of influenza vaccination were lower”, it is misleading to say this Medicaid population since your data source was limited to claims and is not representative over overall vaccination rates since you don’t know the true vaccination rate in the Medicaid population.

o   I wonder if the introduction should be more explicit in the research question of interest. Vaccination rates among those that got vaccinated at a medical facility accepting Medicaid.

Reply: We thank the reviewer for this thoughtful comment. While our data source is limited to Medicaid claims, it is essential to understand that Medicaid captures all vaccinations from pharmacies, employees and vaccination campaigns. This is one of the great benefits of our study population. However, the reviewer correctly points out that some vaccinations administered might still be missed due to missing reporting or errors. Yet, out of the millions of patients included in our population, this number would be negligible. Therefore, we believe that our sentence is correct, however we did add the following sentence to the limitations section to reflect the possibility of errors, based on the Reviewers comment.

“Additionally, although our data source is restricted to Medicaid claims, it is important to emphasize that Medicaid encompasses all vaccinations administered by pharmacies, employees, and vaccination campaigns. Nevertheless, a small number of vaccinations may be missed due to incomplete reporting or errors as well as those that were not considered in this study, such as vaccinations administered at public facilities.

  • “In contrast to other studies that showed higher education associated with seasonal influenza vaccines, this study did not show such association during the study period (Li et al., 2022).” – Li defined education differently than you, this should be stated here.

Reply: We thank the Reviewer for bringing this to our attention. Following the statement noted above, we added the sentence below to the revised manuscript.

“However, it is worth noting that the education data utilized in this study was derived from individuals residing in areas with education levels that were either higher or lower than the median, rendering it inequitable to compare with data acquired from individual self-reports or alternative definitions.”

  • “Our results further suggest that visits for general examination, evaluation, health maintenance encounters, or general mental health encounter is a strong predictor of influenza vaccination for the study period.” – I don’t think your data backs up this statement – did you look at temporality with vaccination? It can only be a predictor if it happened before the vaccine. Maybe vaccination predicts number of visits?

Reply: We thank the Reviewer for identifying this error in our discussion. We have updated the sentence to state: Our results further suggest an association between visits for general examination, evaluation, health maintenance encounters, or general mental health encounters and influenza vaccination during the study period.

Round 2

Reviewer 3 Report

Comments and Suggestions for Authors

The paper can be accepted for publication